# Is breast arterial calcification associated with coronary artery disease?—A systematic review and meta-analysis

Sing Ching Lee[1,2]*, Michael Phillips[3], Jamie Bellinge[1,2], Jennifer Stone[4], Elizabeth Wylie[2,5], Carl Schultz[1,2]

**1** Department of Cardiology, Royal Perth Hospital, Perth, Western Australia, Australia, **2** School of Medicine, University of Western Australia, Perth, Western Australia, **3** Centre for Medical Research (affiliated with the Harry Perkins Institute of Medical Research), University of Western Australia, Perth, Western Australia, Australia, **4** School of Population and Global Health, University of Western Australia, Perth, Western Australia, Australia, **5** Department of Radiology, Royal Perth Hospital, Perth, Western Australia, Australia

* sing.lee@health.wa.gov.au

**Data Availability Statement:** All relevant data are within the manuscript and its Supporting Information files. The protocol is held in a data repository (DOI: 10.26182/5eec2f4221726).

## Abstract

### Background

There is increasing evidence that breast arterial calcification (BAC), an incidental finding on 3–29% of mammograms, could be used to screen for coronary artery disease (CAD). We conducted a systematic review to assess the associations between BAC and CAD and its risk factors (hypertension, hypercholesterolemia, diabetes mellitus and smoking).

### Methods and findings

MEDLINE and EMBASE databases and references of relevant papers were searched up to 18 February 2020 for English language studies that evaluated the associations of BAC and CAD and its risk factors. A single reviewer extracted all data and assessed study quality with verification by another independent reviewer, if required. Across 31 studies (n = 35,583; 3 longitudinal and 28 cross-sectional studies) that examined the association of BAC and CAD, the OR was 2.61 (95% CI 2.12–3.21; $I^2$ = 71%). Sub-analysis of studies that graded BAC severity using the 4- (4 studies) or 12-point scale systems (3 studies) revealed an association with CAD and moderate-severe BAC (OR 4.83 (95%CI 1.50–15.54) and OR 2.95 (95% CI 1.49–5.84), respectively) but not mild BAC (OR 2.04 (95%CI 0.82–5.05) and OR 1.08 (95%CI 0.42–2.75), respectively). BAC was associated with hypertension (42 studies; n = 32,646; OR 1.80; 95% CI 1.47–2.21; $I^2$ = 85%) and diabetes mellitus (51 studies; n = 53,464; OR 2.17; 95% CI 1.82–2.59; $I^2$ = 75%) but not with hypercholesterolemia (OR 1.31; 95%CI 0.97–1.77; $I^2$ = 67%). Smoking was inversely associated with BAC (35 studies; n = 40,002; OR 0.54; 95% CI 0.42–0.70; $I^2$ = 83%). Studies mostly included symptomatic women. Marked heterogeneity existed and publication bias may be present.

### Conclusions

BAC is associated with CAD, diabetes mellitus and hypertension and inversely associated with smoking. Whether BAC could screen for CAD cannot be determined from current published

**Funding:** SC Lee and J Bellinge have been supported by the 'Australian Government Research Training Program Scholarship' from the Commonwealth Government of Australia. The funders had no role in study design, data collection and analysis, decision to publish, or preparation of the manuscript.

**Competing interests:** The authors have declared that no competing interests exist.

data due to the lack of larger prospective studies. A consensus approach to quantifying BAC may also facilitate further translation into clinical care. PROSPERO: CRD42020141644.

## Introduction

Coronary artery disease (CAD) is an important cause of morbidity and mortality globally [1]. Preventative strategies can be effective when targeted at individuals at increased risk of CAD. At present, we rely on cardiovascular risk prediction algorithms to identify persons at high risk of CAD. These probabilistic algorithms often underestimate the risk of CAD in women [2] leading to under prescription of preventative therapies and lifestyle changes. Female-specific strategies may more accurately identify women at risk of CAD.

Coronary calcification is strongly associated with an increased risk of CAD in people without prior CAD events [3]. Non-coronary arterial calcification, such as in the peripheral arteries, is also associated with an increased risk of CAD [4]. It is, therefore, conceivable that detection of non-coronary artery calcification may be useful to screen for quiescent CAD provided that the cost and logistics of screening can be justified.

Mammography is widely used to screen for breast cancer in women aged 40 years and older. Breast arterial calcification (BAC) is a frequent incidental finding on screening mammography but is not routinely reported. It is a form of medial arterial calcification [5] which occur in the absence of atherosclerosis and inflammation [5–7]. Therefore, it has been assumed that BAC or medial arterial calcification are not involved in CAD. Yet, two meta analyses [8,9] and a systematic review [10] have suggested that there may be an association between BAC and CAD and cardiovascular events. This has garnered interest among both clinicians and women who undergo mammography in whether BAC could be used to assess cardiovascular risk in women [11] and in response, there has been a number of publications since these studies have been published that have examined the association between BAC and CAD. The primary aim of this systematic review is to assess the current evidence for the association between BAC and the risk of CAD and whether the association is consistent across different modalities for assessing CAD. The secondary aim of this review is to ascertain if BAC is associated with major cardiovascular risk factors (hypertension, diabetes mellitus, hypercholesterolemia and smoking). We also assessed the effects of the method of quantification of BAC severity on the association with CAD.

## Methods

This review was registered with PROSPERO (CRD42020141644) and was conducted in accordance with the Preferred Reported Items for Systematic Review and Meta-Analyses (PRISMA) statement. The protocol has also been registered (DOI: 10.26182/5eec2f4221726).

### Data sources and searches

A literature search for relevant articles published up to 2 August 2019 was conducted and updated on 18 February 2020 using MEDLINE and EMBASE databases. To provide an updated systematic review of the association of BAC and CAD and its risk factors, a similar search strategy from the previous systematic review [10] was used. This included the following keywords and/or Medical Subject Heading terms: "Breast" OR "mamma" OR "mammary" OR "mammograph*" OR "intramammary" AND "vessel" OR "vessels" OR "artery" OR "arterial" OR "arteries" OR "vascular" AND "calcif*" OR "scleros*" OR "calcinos*" OR "calcium

(S1 Table). Additional papers were identified from the reference lists of papers identified in the search.

## Study selection

Any English language study that examined the associations of BAC and CAD and its risk factors (diabetes mellitus, hypertension, hypercholesterolemia and smoking) were included. To examine CAD specifically, CAD was defined as any cardiovascular events as reported by participants or obtained by medical records (previous myocardial infarction, angina, previous abnormal coronary angiography or functional test, revascularisation) or presence of CAD on coronary angiography (CAG) or presence of perfusion defect on myocardial perfusion scan (MPS) or the presence of atherosclerosis on computed-tomography coronary angiography (CTCA) or CT chest or coronary calcium scores (CCS). When there were studies using the same cohort of participants, only the study of better quality and design (i.e. cohort studies over cross sectional studies) was selected for analysis. Case reports, case series and conference proceedings were excluded as they do not contain sufficient information and were deemed to be at high risk of bias.

All titles and abstracts were screened for inclusion by a single reviewer. Queries regarding the eligibility of a particular study were independently evaluated by a senior reviewer (CS or EW). Any additional references identified from full text review that potentially met the study eligibility criteria underwent the same process as described above.

## Data extraction and quality assessment

Extracted data included: information of the publication (Author, Journal, Year, Title), study design, characteristics of the participants (sample size, average age or age range), prevalence of BAC, description of method of quantifying BAC, description of method of identifying or quantifying CAD (self-report, medical records, CCS, CTCA, CAG or MPS), study's conclusions on the associations of BAC with CAD and its risk factors, record of whether the study accounted for confounders in their final analysis and description of weaknesses of the study. Where available, dichotomous data required to calculate the ORs for the associations of BAC with CAD and its risk factors were also tabulated. When there were multiple information regarding cardiovascular events, objective measures such as abnormal coronary angiography over a history of myocardial infarction was extracted.

Study quality was assessed by a single reviewer (SCL) and quality scores were independently assessed by a second reviewer (CS or EW) where there was uncertainty. The Newcastle Ottawa Scale (NOS) for case control and cohort studies [12] was used to assess the quality of case control and cohort studies, respectively. The NOS uses a nine star point system to evaluate the studies' selection criteria of the study population (four points maximum), method of control for confounders (two points maximum) and the appropriateness of measurement of outcome (three points maximum). To be considered moderate in quality, a study needs to achieve at least two points in the selection criteria, at least one point for controlling for confounders and at least two points in the outcome criteria. A good quality study would need a score of at least three points in the selection criteria, one point for controlling for confounders and at least two points in the outcome criteria. As there is currently no validated measurement tool for assessing the quality of cross-sectional studies, a modified version of the NOS was created to assess the quality of cross sectional studies (S1 Fig). This adapted scale similarly assesses the quality of the studies except a ten point scale system is used (five points for selection criteria; two points for controlling for confounders; three points for outcome criteria). A score of five to six was considered as moderate and a score of seven or more as good in quality.

## Data synthesis and analysis

A random effects model was used for the meta-analysis of BAC and CAD and its risk factors, assuming heterogeneity across different studies. Heterogeneity was tested using the chi-square Cochran's Q test and $I^2$ statistic. Further analysis of the associations between BAC and CAD was conducted by stratifying the studies based on modality used to determine CAD (questionnaire/medical records; CT (CCS, CT chest, CTCA); CAG/MPS) and by method of quantification of BAC (absent/present vs semi-quantitative methods). The association of BAC severity and the presence of CAD was also evaluated. As age is well known determinant of BAC prevalence, a meta-regression analysis was performed with the log odds of the association of BAC with CAD as the dependent variable (y) and the mean difference of age as an independent variable (x). The mean difference of age was calculated by subtracting the average mean age of all the studies included in the analysis from the mean age of each study. Where a mean age was not provided, the median value was used. Study quality was also investigated as an independent variable (x). Sensitivity analyses excluding studies with less than 100 participants was performed to assess the impact on heterogeneity. The risk of publication bias in studies that have assessed the association of BAC with CAD was evaluated using a funnel plot. For the examination of the associations of BAC and cardiovascular risk factors, subgroup analyses involving only good quality studies were performed for each variable to assess if study quality was a determinant of the heterogeneity between studies. A p-value of less than 0.05 was considered to be statistically significant. Analyses were performed using R studio (version 3.5.2) and the 'meta' and 'metafor' packages.

## Results

A total of 3,724 scientific publications were identified through database searching. After removal of duplicates and hand searching references of these articles, there were 2,930 unique publications of which 205 were original research articles that were suitable to be retrieved for full text assessment for eligibility. Of these, only 58 articles were deemed relevant and one additional study was identified on an updated searched which resulted in a total of 59 studies that were included in our final analysis (Fig 1). These were all cross-sectional observational studies apart from five that were longitudinal studies. A total of 26 studies were considered good in quality. The characteristics of these 59 studies are provided in Table 1 and the critical appraisals are provided in S2–S4 Tables.

### Studies evaluating BAC and CAD

Of the 59 studies, 38 examined the association between BAC and CAD. Out of these, only 31 studies (n = 35,583) had data available to calculate the OR of the association of BAC and CAD. Only three of these studies were longitudinal and the rest were cross sectional. Only four studies reported data on women with no prior history of CAD. Most of these studies (n = 10) assessed the presence of CAD using questionnaires or medical records. Two studies used CT chest, three studies used CCS, five studies used CTCA and 10 used CAG and one used MPS to assess the presence of CAD. Furthermore, of the 31 studies, 19 studies assessed BAC by its presence/absence and 12 studies visually graded the severity of BAC. For studies that graded BAC severity, five used the same four-point scale system, three used the same 12-point scale system with the remaining four studies using scaling systems that were different from one other (Table 2). Out of these, only 9 studies had data available to calculate the OR of the association of mild BAC or moderate to severe BAC and the presence of CAD.

**BAC and CAD.** The pooled OR of the association of BAC and CAD was significant at 2.61 (95% CI 2.12–3.21). Heterogeneity across studies was significant at $I^2$ 71% (p<0.01)

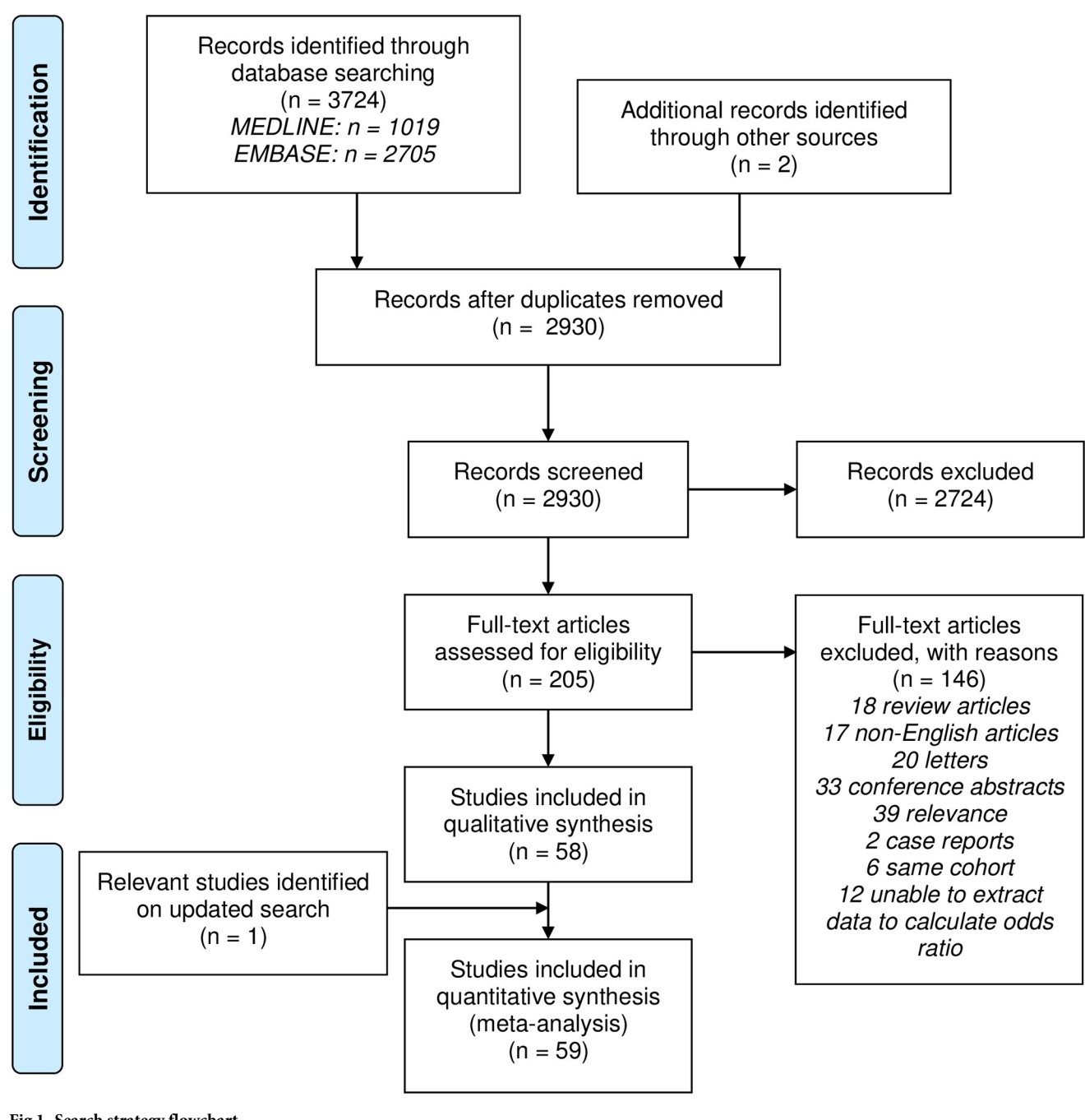

**Fig 1. Search strategy flowchart.**

(Fig 2A). When only studies of women with no prior history of CAD were included, the pooled OR of the association of BAC with the presence of CAD was 3.46 (95% CI 1.57–7.61). When studies were analysed based on modality of determining CAD, the OR were 2.16 (95% CI 1.83 to 2.55), 2.14 (95% CI 1.40 to 3.29) and 3.90 (95% CI 2.53 to 6.03) for questionnaires, CAG/ MPS and CT, respectively (Fig 2B). There was no heterogeneity between studies that examined CAD based on questionnaires ($I^2$ = 15%; p = 0.30) whereas heterogeneity between studies that

**Table 1. Characteristics of studies included in the meta-analysis.**

| Author | Year | Design | Follow up period | N | Age (mean) | BAC prevalence |
|---|---|---|---|---|---|---|
| Baum [13] | 1980 | CS | - | 319 | ? | 11.6% |
| Sickles [14] | 1985 | CS | - | 5000 | ? | 9.6% |
| Van Noord [15] | 1996 | CS | | 12,239 | (57.7) | 9.0% |
| Crystal [16] | 2000 | CS | - | 865 | (65/54 BAC+/-) | 17.6% |
| Cetin [17] | 2004 | CS | - | 2400 | 32–75 | 9.1% |
| Iribarren [18] | 2004 | Cohort | 25 years | 12,761 | 40–79 | 3.0% |
| Maas (I) [19] | 2006 | CS | - | 1699 | 49–70 | 11.4% |
| Taşkin [20] | 2006 | CC (BAC) | - | 985 | >40 | 7.9% |
| Dale (II) [21] | 2008 | CS | - | 1000 | 23–93 | 16.0% |
| Pidal [22] | 2009 | CC (BAC) | - | 136 | (57/55 BAC+/-) | 8.4% |
| Akinola [23] | 2011 | CS | - | 54 | (60/51 BAC+/-) | 20.0% |
| Sedighi [24] | 2011 | CC (BAC) | - | 204 | (60.6) | 14.7% |
| Schnatz [25] | 2011 | Cohort | 5 years | 1,454 | (56.3) | 14.2% |
| Friedlander [26] | 2012 | CS | - | 40 | (62.2) | 22.5% |
| Zafar [27] | 2013 | CS | - | 200 | (56/44 BAC+/-) | 13.5% |
| Kosovic [28] | 2015 | CC (BAC) | - | 300 | (62.0) | ? |
| Hanafi [29] | 2018 | CC (CAD) | - | 60 | (53/51 CAD+/-) | 60.0% |
| Soylu [30] | 2018 | CS | - | 404 | >40 | 30.4% |
| Bae [31] | 2013 | CC (BAC) | - | 201 | (58.9) | - |
| Matsumura [32] | 2013 | CC (BAC) | - | 202 | (60/58 BAC+/-) | - |
| Yagtu [33] | 2015 | CC (BAC) | - | 80 | 39–86 | ? |
| Margolies [34] | 2016 | CS | - | 292 | (61.5) | 42.5% |
| Moshyedi [35] | 1995 | CS | - | 182 | 39–92 | 24.2% |
| Henkin [36] | 2003 | CC (CAD) | - | 319 | (61.8) | 41.1% |
| Fiuza Ferreira [37] | 2007 | CS | - | 131 | (61.1) | 39.7% |
| Topal [38] | 2007 | CS | - | 123 | (64/52 BAC+/-) | 39.8% |
| Oliveira [39] | 2009 | CC (CAD) | - | 80 | (64.65) | - |
| Penugonda [40] | 2010 | CS | - | 94 | (66.7) | 60.6% |
| Zgheib [41] | 2010 | CC (CAD) | - | 172 | (64.3) | 33.1% |
| Hekimoğlu [42] | 2012 | CS | - | 55 | (63) | 41.8% |
| Moradi [43] | 2014 | CS | - | 150 | (68/54 BAC+/-) | 23.3% |
| Karm [44] | 2015 | CS | - | 198 | (65) | 41.4% |
| Chadashvili [45] | 2016 | CS | - | 145 | (56/61 BAC+/-) | 25.5% |
| Fathala (I) [46] | 2017 | CS | - | 435 | (58) | 59% |
| Kelly [47] | 2018 | Cohort | 20.6 months | 104 | (58.93) | 14% |
| Ružičić [48] | 2018 | CS | - | 102 | (62) | 63.7% |
| McLenachan [49] | 2019 | CS | - | 405 | (58) | 23.0% |
| **Population: post-menopausal** | | | | | | |
| Kataoka [50] | 2006 | CS | - | 1590 | (63.2) | 16.0% |
| Yildiz (III) [51] | 2008 | CC (BAC) | - | 54 | (62.7) | 10.2% |
| Ferreira [52] | 2009 | CS | - | 307 | (55.3) | 8.5% |
| Nasser [53] | 2014 | CS | - | 211 | (62.1) | 18.0% |
| Atci [54] | 2015 | CC (BAC) | - | 567 | (65/55 BAC+/-) | 31.6% |
| Pecchi [55] | 2003 | CS | - | 74 | <65 | 59.5% |
| Yildiz (II) [56] | 2014 | CS | - | 310 | (55.9) | 33.9% |
| Parikh [57] | 2019 | CS | - | 3507 | 60–79 | 27.9% |
| Maas (II) [58] | 2004 | CS | - | 600 | (70/67 BAC+/-) | 23.0% |

*(Continued)*

**Table 1.** (Continued)

| Author | Year | Design | Follow up period | N | Age (mean) | BAC prevalence |
|--------|------|--------|------------------|---|------------|----------------|
| *Population: diabetes* | | | | | | |
| Schmitt [59] | 1985 | CC (DM) | - | 450 | 35–74 | 45.1% |
| Dale (I) [60] | 2010 | CC (DM) | - | 1609 | 24–93 | 36.5% |
| Sankaran [61] | 2019 | CC (BAC) | - | 100 | (59/51 BAC+/-) | ? |
| *Population: chronic kidney disease* | | | | | | |
| Abou Hassan [62] | 2015 | Cohort | 3.3 years | 202 | (61/ 54 BAC+/-) | 58.4% |
| Voyvoda [63] | 2019 | CS | - | 55 | (54.8) | 14.5% |
| *Other* | | | | | | |
| Wada [64] | 2012 | CC (breast cancer) | - | 3771 | (56/58 breast cancer +/-) | 9.9% (breast cancer) 14.3% (no breast cancer) |
| Soran [65] | 2014 | Cohort (breast cancer) | 7.5 years | 602 | (62/54 BAC+/-) | 26.7% |
| Mostafavi [66] | 2015 | CS | - | 100 | (65.3) | 12.0% |
| Yildiz (IV) [67] | 2016 | CC (premenopausal) | - | 166 | (45/45 BAC+/-) | ? |
| Ronzani [68] | 2017 | CS | - | 312 | (55.9) | 23.0% |
| Fathala (II) [69] | 2018 | CS | - | 307 | (54/59 BAC+/-) | 46.3% |
| Yildiz (I) [70] | 2018 | CC (BAC) | - | 132 | (54) | - |
| Sarrafzadegann [71] | 2009 | CS (premenopausal) | - | 84 | <55 | 7.1% |

Abbreviations—CC: case control (variable used to separate cases and controls); BAC: breast arterial calcification; CS: cross sectional; DM: diabetes mellitus; CAD: coronary artery disease; CKD: chronic kidney disease; Green shading indicates studies that only included participants with no known history of cardiovascular disease; Orange shading indicates studies that only included participants with known history of cardiovascular disease.

examined CAD based on CAG/MPS and questionnaires remained significant at $I^2$ = 74% (p<0.01) and $I^2$ = 75% (p<0.01), respectively.

When studies were stratified based on method of reporting BAC, the OR were 2.05 (95% CI 1.70–2.47) and 4.04 (95% CI 2.59–6.30) for studies that reported BAC by presence/absence and semi-quantitatively, respectively, with no changes to heterogeneity. A sub-analysis was performed on studies that used the 4-point (n = 4 studies) and 12-point scale system (n = 3 studies). For those that used the 4 point scale system, mild BAC had a pooled OR of 2.04 (95% CI 0.82–5.05) (Fig 3A) and moderate to severe BAC had a pooled OR of 4.83 (95% CI 1.50–15.53) (Fig 3B). For those that used the 12 point scale system, the pooled OR was 1.08 (95% CI 0.42–2.75) for mild BAC and 2.95 (95% CI 1.49–5.84) for moderate to severe BAC (Fig 3C and 3D).

Study quality did not have a significant impact on the association of BAC and CAD (Slope = 0.11, 95% CI -0.79 to 1.01, p = 0.82) (S2 Fig). Interestingly, the ORs decreased with age, but this was not statistically significant (Slope = -0.03, 95%CI -0.10 to 0.04, p = 0.37) (S3 Fig). Sensitivity analysis excluding studies with less than 100 participants did not have a large impact on heterogeneity ($I^2$ 75% to 69%). The funnel plot was also asymmetrical (Egger's test p value < 0.001) (S4 Fig).

## Studies evaluating BAC and cardiovascular risk factors

Out of the relevant 59 articles, respectively 42 (n = 32,646), 51 (n = 53,464), 27 (n = 11,652) and 35 (n = 40,002) studies had data available that enabled calculation of ORs for the associations of BAC with hypertension, diabetes mellitus, hypercholesterolemia and smoking. Of these, the number of studies considered good in quality were 19 that reported on hypertension, 21 that reported on diabetes mellitus, 13 that reported on hypercholesterolemia and 20 that reported on smoking.

**Table 2. Studies that have visually graded BAC severity and examined the association of BAC severity and CAD.**

| Author | Scale used | Scale definition |
|---|---|---|
| Mostafavi [66], Hanafi [29], Ružičić [48], McLenachan[49], Soylu [30] | 4 point scale | 1: no vascular calcification |
| | | 2: few punctate vascular calcifications, no areas of tram track or ring calcifications (mild) |
| | | 3: coarse vascular calcifications of definite tram track or ring appearance affecting <3 vessels (moderate) |
| | | 4: severe coarse vascular calcifications affecting >3 vessels (severe) |
| Pecchi [55] | 3 scale; uni/bilateral breast | Mild: only isolated calcified plaques along the course of the vessels |
| | | Moderate: typical parallel track calcifications seen along part of the arterial vessels |
| | | Severe: entire, 'punched out' calcified vessels |
| Oliviera [39] | 4 point scale | Absent: no vascular calcification |
| | | Slight: arteries clearly outlined by calcification with distances greater than 10mm between calcified areas |
| | | Moderate: arteries clearly outlined by calcifications over a considerable proportion of their course |
| | | Severe: arteries extensively affected, seen to have almost continuous columns of calcification with at least two branches also visible |
| Moradi [43] | 4 point scale | Normal: no calcification |
| | | Mild: slightly calcified breast artery |
| | | Moderate: distinctly calcified breast artery |
| | | Severe: solid calcification of the breast artery |
| Topal [38] | Uni/bilateral breast; number of blood vessels involved; continuous or discontinuous calcifications | - |
| Margolies [34], Fathala (I) [46], Fathala (II) [69] | 12 point scale (sum of scores for density, length and number of vessels involved | Density of calcium in most affected segment (0: none; 1: mild with clear visualisation of the lumen and/or only one vessel involved; 2: clouding of lumen and calcification of both tangential walls; 3: severe with no visible lumen) |
| | | Longest length of vessel involvement (0: none; 1: less than one third; 2: one third to two thirds; 3: more than two thirds of vessel length |
| | | Number of vessels (1 to 6) |
| | | Mild: score 1 to 4 |
| | | Moderate: score 5 to 8 |
| | | Severe: score 9 to 12 |

**BAC and cardiovascular risk factors.** Hypertension (OR 1.80; 95% CI 1.47–2.21), diabetes mellitus (OR 2.17; 95% CI 1.82–2.59) and hypercholesterolemia (OR 1.28; 95% CI 1.06–1.55) were significantly associated with the risk of BAC. In contrast, smoking was associated with approximately half the risk for having BAC compared to non-smokers (OR 0.54; 95% CI 0.42–0.70). Marked heterogeneity existed across studies for all analyses (Fig 4).

When only good quality studies were included in the analysis, the ORs for hypertension increased to 1.99 (95% CI 1.59–2.49) and the ORs for diabetes mellitus and smoking decreased to 1.76 (95% CI 1.46–2.12), and 0.51 (95% CI 0.37–0.69), respectively, whereas the OR for hypercholesterolemia became insignificant (OR 1.31 (95% CI 0.97–1.77). Heterogeneity remained significant.

## Discussion

Our quantitative review of the literature showed a positive associations of BAC with risk of CAD and with most of the major cardiovascular risk factors except for smoking. Sub-analysis

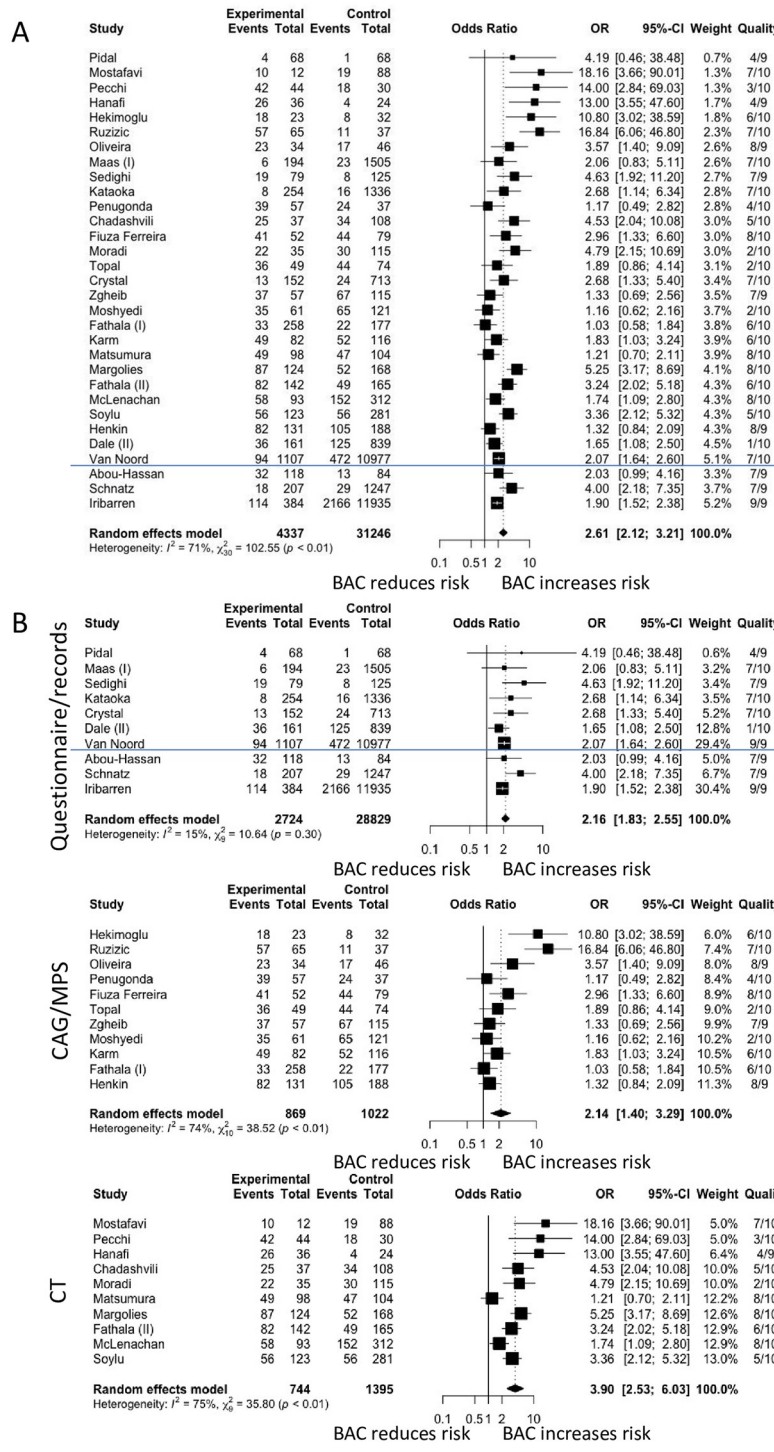

**Fig 2. Forest plot analysis of breast arterial calcification (BAC) and coronary artery disease (CAD).** Fig 2A shows the association of BAC and CAD when stratified based on study design. Fig 2B shows the association of BAC and CAD when stratified based on modality of determining CAD. The blue line separates cross sectional and cohort studies. (CT: computed tomography; CAG: coronary angiography.

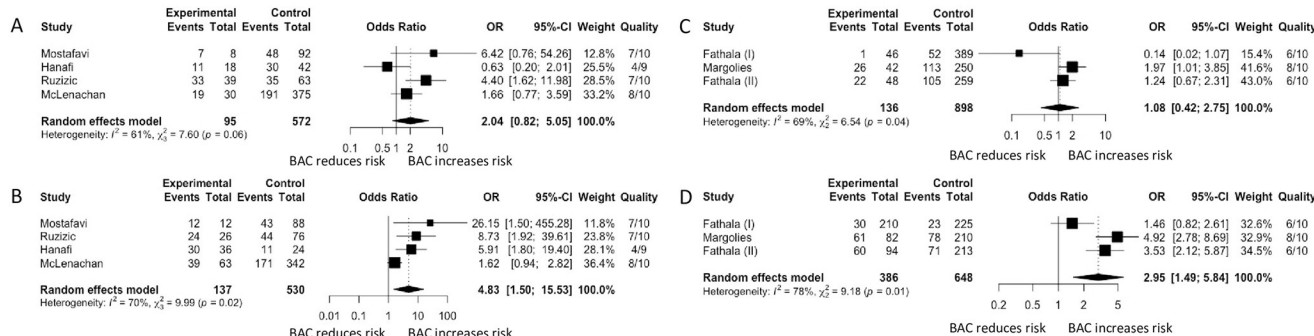

**Fig 3. Forest plot analysis of breast arterial calcification (BAC) and coronary artery disease (CAD) based on BAC severity.** Fig 3A and 3B shows the association of BAC and CAD using the 4 point scale system. Fig 3C and 3D shows the association of BAC and CAD using the 12 point scale system. Fig 3A and 3C are the forest plot analysis for mild BAC and CAD and Fig 3B and 3D for moderate to severe BAC and CAD.

on BAC severity and the association of CAD revealed that only moderate to severe BAC was associated with CAD. There was marked heterogeneity across studies in all our analyses. The majority of studies were cross-sectional in design and included symptomatic patients which limits interpretation.

## BAC and CAD

We report a positive association between BAC and CAD with a summarised OR of 2.61 (95% CI 2.12–3.21). This is consistent with a previous systematic review that identified only three longitudinal studies and which reported a HR of 1.32 to 1.44 for the association between BAC and risk of CAD [10]. However, in these studies, CAD was retrospectively assessed from medical history or records. Two other meta-analysis by Abi Rafeh et al [8]and Jiang et al [9] that included only studies that assessed CAD with CAG reported that BAC was positively associated with the risk of CAD with an OR of 1.59 (95%CI 1.21–2.09) and 3.86 (95%CI 3.24–4.59), respectively. However, the populations studied were highly selected because CAG is an invasive investigation that is almost exclusively performed on patients with symptoms thought to be due to CAD. Therefore, conclusions cannot be drawn on the usefulness of BAC for screening which, by definition, concerns asymptomatic populations without prior cardiovascular events or cardiovascular symptoms. Furthermore, CAG underestimates the prevalence of CAD because luminal narrowing may not be seen in the presence of arterial remodelling even if the atherosclerotic plaque burden is very high. We attempted to address the limitations of prior meta-analyses by including all studies, irrespective of method of assessment of CAD, resulting in a much higher number of studies The analysis were further stratified by method of assessment of CAD.

When we stratified our analysis based on modality of determining CAD, the association of BAC and CAD identified by CT was the strongest among the three modality groups (OR = 3.90). Assessment of CAD by questionnaires are well recognised to be open to recall and other biases, while CAG can only detect luminal stenosis and may miss the presence of quiescent CAD. Of all the methods used to define CAD in published studies, CT is the most accurate method of detecting the presence of any CAD. These data suggest that BAC could potentially be used to screen for CAD. However, our review identified a major limitation in that most of the studies did not exclude women with prior cardiovascular events, which limits inferences on using BAC for screening purposes. We identified only four studies that included women without CAD (total n = 875). All showed a positive association between BAC and CAD with a pooled OR of 3.46 (95% CI 1.57–7.61). The funnel plot was also asymmetrical

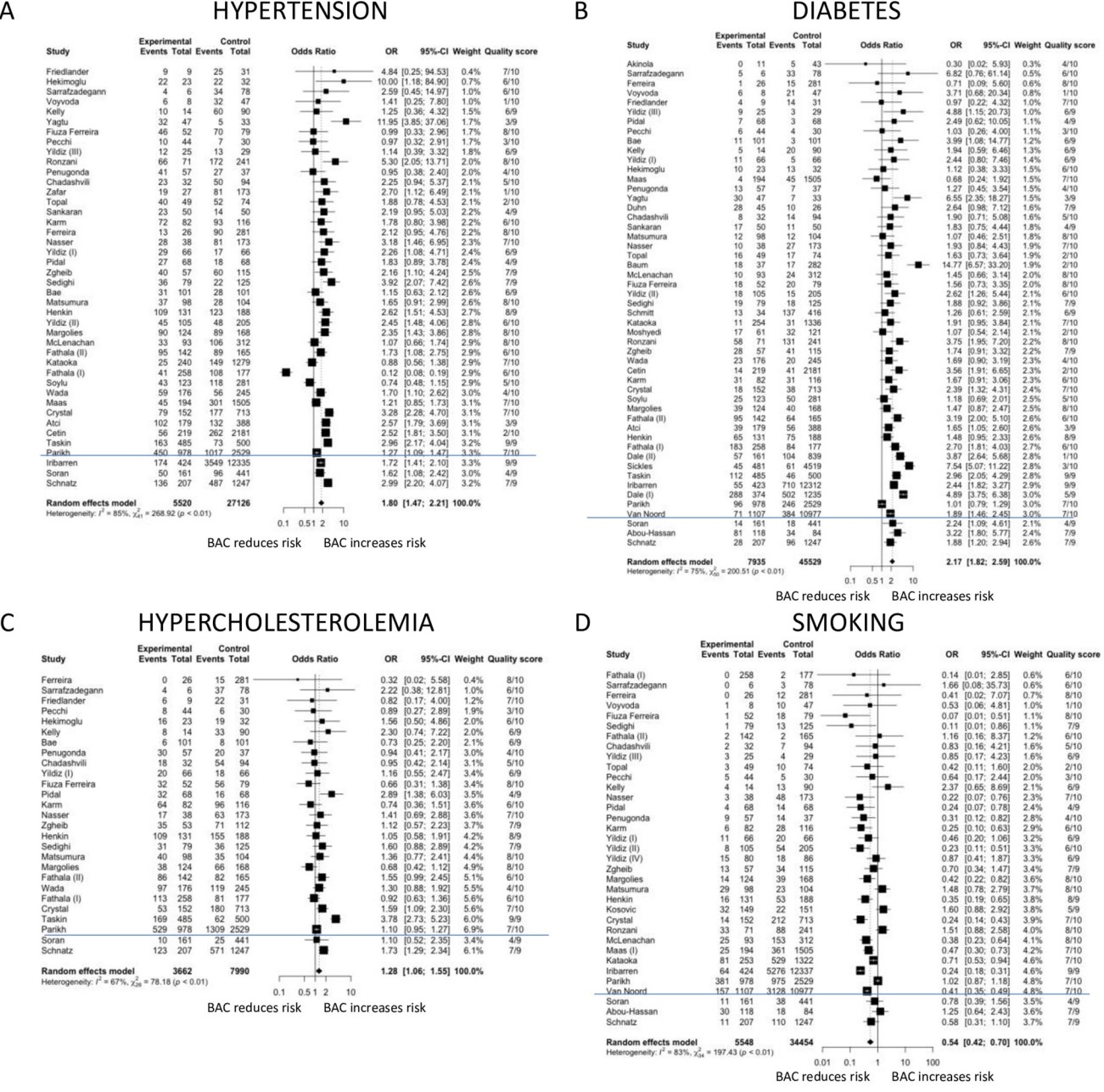

**Fig 4. Forest plots of associations of breast arterial calcification with cardiovascular risk factors.** Odds ratios are shown for studies reporting data on hypertension (Fig 4A), diabetes mellitus (Fig 4B), hypercholesterolemia (Fig 4C) and smoking (Fig 4D). The blue line separates cross sectional and cohort studies.

suggesting the potential presence of publication bias. Therefore, there is insufficient data to determine whether BAC may be used as a marker for undeclared CAD. Larger prospective studies involving women with no prior history of CAD at baseline are required to determine whether incidental BAC reliably detects quiescent CAD.

There was significant heterogeneity across all the studies in our analysis. When our analysis was stratified based on method of determining CAD, heterogeneity remained significant for studies that assessed CAD by CAG, CT or MPS but not questionnaires. This was in contrast to

a previous systematic review that only included studies that looked at CAD assessed by CAG [9]. We assessed whether study quality contributed to heterogeneity by performing a meta-regression analysis, but found that it did not. A sensitivity analysis, excluding small studies with participant numbers of less than 100 also did not significantly alter the effects of heterogeneity. We also excluded age as a significant confounder or contributor to heterogeneity which reinforces the suggestion that BAC may be independently associated with CAD. We were unable to control for variations in other characteristics between these studies, such as the different methods of quantifying outcomes that may have contributed to the heterogeneity found across all the studies. Yet, the heterogeneity may also indicate that the overall association between BAC and the presence of CAD is widely generalizable [72].

We also found no changes in the association of BAC and CAD and heterogeneity regardless of how BAC was assessed (presence/absence: OR 2.05 (95% CI 1.70–2.47); semi-quantified: OR 4.04 (95% CI 2.59–6.30)). The higher ORs seen in the semi-quantitative method are most likely explained by population differences. Alternatively, it may suggest that semi-quantitative methods are more sensitive in detecting subtle forms of BAC but a head-to-head comparison is required to confirm this. Interestingly, we also observed a significant association between moderate to severe BAC and CAD (OR 4.83 (95% CI 1.50–15.53) and OR 2.95 (95% CI 1.49–5.84) for 4 point scale and 12 point scale, respectively) but not with mild BAC (OR 2.05 (95% CI 0.82–5.05) and OR 1.08 (95% CI 0.42–2.75) for 4 point scale and 12 point scale, respectively). This lack of association between mild BAC and CAD could be explained by the lack of power in these studies to detect a possible association but overall, suggest that there may be an important association between BAC severity and CAD. Additional work using a grading system in longitudinal studies are required. However, there is currently no consensus on a method of assessing BAC. We are unable to comment on which method of assessing BAC is superior over the other as this requires a dedicated study. Furthermore, these methods are semi-quantitative and may be more subjective. A fully quantitative method of assessing BAC may be more objective.

## BAC and cardiovascular risk factors

We report ORs describing the risk of BAC associated with diabetes mellitus and hypertension of 2.17 (95%CI 1.82–2.59) and 1.80 (95%CI 1.47–2.21), respectively. We also initially found that BAC was associated with the presence of hypercholesterolemia, which is a risk factor for intimal arterial calcification or atherosclerosis [73], with an OR of 1.28 (95%CI 1.06–1.55) but this became insignificant when only good studies were included with an OR of 1.31 (95%CI 0.97–1.77). A previous systematic review [10] reported only an association between BAC and diabetes mellitus but included far fewer studies and patients and consequently may have lacked the statistical power to identify the association between BAC and hypertension. As BAC is a form of medial arterial calcification [5], our findings are consistent with well-known associations of medial arterial calcification for diabetes mellitus [73]. The association of BAC and hypertension is interesting as hypertension can promote atherosclerosis and calcification within the tunica media which leads to arterial wall stiffening which in turn could contribute to the development of hypertension [74], causing a vicious cycle between hypertension and vascular calcification.

Smoking was found to be inversely associated with the risk of BAC in our study with an OR of 0.54 (95%CI 0.42–0.70). This was similarly observed in a previous systematic review [10]. Yet, it is recognised that coronary artery calcification risk is increased with smoking [75]. The reasons for this apparent paradox are not known. Potential selective survival of smokers without BAC may in part account for this but another explanation could be that smoking may be a

marker of an unidentified confounder. Several studies have attempted to control for smoking status through logistic regression and still reported a positive association between BAC and CAD [32,34,62,66]. However, no published studies have stratified their findings between BAC and CAD based on smoking status, which may lead to a different interpretation of the association of BAC and CAD.

## Limitations

A major limitation of this systematic review is that the results are based on published data which have major limitations. For example, most of the studies included in our analysis were cross sectional in design and only a few of the studies specifically targeted women with no prior history of CAD. Therefore, current evidence is insufficient to determine if the measurement of BAC could be used as a screening tool for quiescent CAD. At present, there is also no validated tool to grade the quality of cross sectional studies. We have attempted to overcome this by modifying a standardised method used to appraise cohort and case control studies and which has been used in previous systematic reviews [76], but this approach requires validation. We also demonstrated the possibility of publication bias in our study using a forest plot, however, the marked heterogeneity across all the studies could also have accounted for the asymmetry observed on our funnel plot analysis. Another limitation of this systematic review was that bias may be introduced as only one reviewer was primarily involved in selecting, assessing and extracting data from the studies included in this analysis. However, there were secondary reviewers who were available if there were any concerns regarding any of the study included. As our primary outcome included all studies regardless of quality, it is also unlikely that our conclusions would have changed. Finally, we analysed only studies where published data were available.

## Conclusion

BAC is associated with CAD and hypertension and diabetes mellitus, but is inversely associated with smoking. However, as most of these studies were cross sectional and included symptomatic women, it remains unclear whether the measurement of BAC could be used as a screening tool for CAD in asymptomatic women who undergo mammography. There may also be an important association between BAC severity and CAD. A consensus on the optimum approach to grade or quantitate BAC is required. Future studies should exclude subjects with known CAD or prior cardiovascular events and ideally use a widely accepted grading system for BAC and be prospective to also enable assessment of the predictive value of BAC and its severity for cardiovascular events.

## Supporting information

**S1 Table. Search strategy used in MEDLINE.** * Asterisk used to capture multiple word endings (e.g. calcifying, calcification).
(DOCX)

**S2 Table. Critical appraisal of cohort studies included in the meta-analysis.** 1: representativeness of cohort (a—truly representative of the community; b—somewhat representative of the community; c—selected group; d—no description), 2: selection of non-exposed cohort (a—drawn from the same community as the exposed cohort; b—drawn from a different source; c—no description), 3: ascertainment of exposure (a—secure record or independent blind assessment; b—single blinded assessment; c—unblinded assessment; d—no description), 4: demonstration that outcome of interest was not present at the start of study (a—yes; b—no), 5: comparability/controlling for confounders (a—study controls for age; b—study controls for

any additional factor), 6: assessment of outcome (a—independent blind assessment; b—record linkage; c—self report; d—no description), 7: duration of follow up (a—yes; b—no), 8: adequacy of follow up (a—complete follow up; b—subjects lost to follow up unlikely to introduce bias where >80% were followed up or description was provided of those lost; c—follow up rate <80% and no description of those lost; d—no statement), *: stars given for each question; x: answer marked for each question, +: good, ±: moderate.
(DOCX)

**S3 Table. Critical appraisal of case control studies included in the meta-analysis.** 1: case definition (a—yes with independent validation; b—yes with record linkage, self report, non-independent validation; c—no description), 2: representativeness of cases (a—consecutive or obviously representative series of cases; b—potential selection bias or not stated), 3: selection of control (a—community controls; b—hospital controls; c—no description), 4: definition of control (a—no history of disease/endpoint; b—no description), 5: comparability/adjusting for confounders (a—controls for age; b—controls for other additional factors), 6: ascertainment of exposure (a—secure record or two blinded investigators; b—structured interview where blinded to case/control status or one blinded investigator; c—not blinded; d—written self report or medical record only; e—no description), 7: same method of ascertainment of exposure for cases and controls (a—yes; b—no), 8: non response rates (a—same rate for both groups; b—non respondents described; c—rate different and no designation), *: stars given for each question; x: answer marked for each question; +: good, ±: moderate.
(DOCX)

**S4 Table. Critical appraisal of cross sectional studies included in the meta-analysis.** 1: representativeness of the sample (a—truly representative of the average target population; b—somewhat representative of the average target population; c—selected group of participants; d—no description); 2: sample size (a—justified and satisfactory including sample size calculation; b—not justified or no information), 3: Ascertainment of risk factor (a—secured record/clinical registry and validated measurement tool used; b—structured questionnaire and validated measurement tool; c—self report; d—no description), 4: non-respondents/missing comparable data (a—the characteristics between BAC+/- are comparable and response rate is >50%; b—the characteristics between BAC+/- is not sufficiently comparable and the response rate is <50%; c—no information provided), 5: comparability/control for confounders (a—data/results adjusted for age; b—data/results adjusted for any additional factors), 6: assessment of BAC (a—independent blinded assessment of BAC; b—single blinded assessment of BAC; c—unblinded assessment of BAC/no description, 7: statistical methods (a—statistical test used to analyse the data clearly described, appropriate and measures of association presented include confidence intervals and p values; b—statistical test not appropriate, not described or incomplete), *: stars given for each question; x: answer marked for each question, +: good, ±: moderate.
(DOCX)

**S1 Fig. Newcastle Ottawa Scale used to assess the quality of cross-sectional studies.**
(TIF)

**S2 Fig. Meta-regression plot of the odds of breast arterial calcification (BAC) being associated with coronary artery disease (CAD) against the quality score of the study.** Each bubble represents one single study and its size corresponds to the study sample size. The odds of BAC being associated with CAD had a slight upward trend with quality of the study (slope = 0.059; p = 0.889).
(TIF)

**S3 Fig. Meta-regression plot of the odds of breast arterial calcification (BAC) being associated with coronary artery disease (CAD) against standardised mean age.** Standardised mean age was calculated as mean age of all study included in analysis—mean age of individual study. Each bubble represents one single study and its size corresponds to the study sample size. The odds of BAC being associated with CAD had a downward trend with age (slope = -0.032, p = 0.323).
(TIF)

**S4 Fig. Funnel plot of studies that examined the association between breast arterial calcification (BAC) and coronary artery disease (CAD).** Each dot represents a study. The solid vertical line represents the summary estimate of the log odds ratio of BAC and CAD and the triangle is fixed on the summary estimate and extends 1.96 standard errors either side. The distribution of studies are asymmetrical and more than 5% of the studies are outside the white triangle which indicates potential publication bias but could also result from heterogeneity.
(TIF)

**S5 Fig. Forest plot analysis of breast arterial calcification (BAC) and coronary artery disease (CAD) based on study design.** S5 Fig A shows the association of BAC and CAD in cross-sectional studies. S5 Fig B shows the association of BAC and CAD in cohort studies.
(TIF)

**S1 Checklist. PRISMA 2007 checklist.**
(DOC)

## Author Contributions

**Conceptualization:** Sing Ching Lee, Jennifer Stone, Elizabeth Wylie, Carl Schultz.

**Data curation:** Sing Ching Lee.

**Formal analysis:** Sing Ching Lee, Michael Phillips.

**Investigation:** Sing Ching Lee.

**Methodology:** Sing Ching Lee, Michael Phillips, Jennifer Stone, Carl Schultz.

**Supervision:** Jennifer Stone, Elizabeth Wylie, Carl Schultz.

**Validation:** Carl Schultz.

**Writing – original draft:** Sing Ching Lee.

**Writing – review & editing:** Sing Ching Lee, Michael Phillips, Jamie Bellinge, Jennifer Stone, Elizabeth Wylie, Carl Schultz.

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
