## [Decision Letter · Decision Letter 0]

21 May 2020

PONE-D-20-11782

Is breast arterial calcification associated with coronary artery disease? - A systematic review and meta-analysis

PLOS ONE

Dear Dr Lee,

Thank you for submitting your manuscript to PLOS ONE. After careful consideration, we feel that it has merit but does not fully meet PLOS ONE’s publication criteria as it currently stands. Therefore, we invite you to submit a revised version of the manuscript that addresses the points raised during the review process.

We would appreciate receiving your revised manuscript by Jul 05 2020 11:59PM. To enhance the reproducibility of your results, we recommend that if applicable you deposit your laboratory protocols in protocols.io, where a protocol can be assigned its own identifier (DOI) such that it can be cited independently in the future. For instructions see: http://journals.plos.org/plosone/s/submission-guidelines#loc-laboratory-protocols

We look forward to receiving your revised manuscript.

Kind regards,

Pascal A. T. Baltzer, M.D.

Academic Editor

PLOS ONE

Journal Requirements:

"The authors, SC Lee and J Bellinge, were supported by an ‘Australian Government Research Training Program Scholarship’ from the Commonwealth Government of Australia."

"The author(s) received no specific funding for this work"

Reviewers' comments:

Reviewer's Responses to Questions

**Comments to the Author**

1. Is the manuscript technically sound, and do the data support the conclusions?

Reviewer #1: Yes

Reviewer #2: Yes

2. Has the statistical analysis been performed appropriately and rigorously? 

Reviewer #1: Yes

Reviewer #2: Yes

3. Have the authors made all data underlying the findings in their manuscript fully available?

Reviewer #1: No

Reviewer #2: Yes

4. Is the manuscript presented in an intelligible fashion and written in standard English?

Reviewer #1: No

Reviewer #2: Yes

5. Review Comments to the Author

Reviewer #1: In this paper, the authors aim to assess the associations between breast arterial calcification (BAC) and the risk of cardiovascular disease and its risk factors (hypertension, hypercholesterolemia, diabetes mellitus and smoking). A systematic literature search updated until February 2020 was conducted and a meta-analysis was finally performed on 59 good-quality studies according to modified Newcastle Ottawa Scale (NOS) scale. Odds ratios and 95% confidence intervals were extracted that examined risk factors for BAC or associations of BAC with cardiovascular disease. Age was taken into account as a confounding variable and meta-regression analysis was performed. Heterogeneity among studies was verified. No publication bias was encountered. Conclusions state that BAC is associated with CAD and hypertension and diabetes mellitus, but is inversely associated with smoking.

It remains unclear whether the measurement of BAC could be clinically translated and future studies are needed.

Strengths

− A good methodology was applied in systematic review and meta-analysis.

− No publications bias occurred.

− Association of BAC with cardiovascular risk factors and disease was confirmed

− Sections are consistent throughout the manuscript and adequately written and organized

Weaknesses

− The current meta-analysis does not add to the body of evidence concerning BAC and cardiovascular risk.

− Search strategy should be explicated in Methods.

− Even if this review was conducted in concordance with the PRISMA statement, ideally two researchers should independently perform the data extraction.

− As description of method of quantifying BAC has been extracted, it would have been interesting to present results about that, that currently represent the crucial gap in clinical translation of BAC for preventive cardiology in women. No published systematic data are on this.

− What’s the BAC prevalence estimate?

− Table 1 report only 50 studies. Are 9 studies missing?

− What’s the EMBED database? Do you mean EMBASE? Otherwise please give source.

Conclusion

− The literature search and meta-analysis are well conducted

− The paper is well written but does not add in evidence.

− Discussion could be enriched

Reviewer #2: This study is a systematic review and meta-analysis for studies of BAC as risk factor of cardiovascular disease and its risk factors. The results show that BAC is indeed associated with CAD, diabetes and hypertension

Several comments can be made:

1. Introduction: no mention is made of the fact that BAC concerns a specific form of calcification namely medial calcification and that this form of calcification may also have different pathways to lead to CVD. This should be indicated in the paper.

2. Introduction: an earlier systematic review and meta-analysis was performed on this topic (Hendriks EJ, atherosclerosis, 2015). This study should be referenced and the authors should indicate what this additional meta-analysis will add.

3. Methods; only 1 reviewer performed study selection, data extraction and quality assessment. This is a limitation of the study.

4. Methods: can the authors provide a little more detail on how quality was assessed for study population, control for confounding and the outcome. The number of points is indicated but not how this is specifically applied to these studies and which items under these headings were scored. This information could be provided in a supplement.

5. Methods; CAD is very heterogenous endpoint. Can the authors perform subgroup analyses for different outcomes such as coronary heart disease and stroke or otherwise specify wheter only events or also CAD mortality was investigated and perform subgroup analyses for this.

6. Results: can the authors also do subgroup analysis separating cross-sectional from prospective studies?

7. Discussion: the authors indicate that only 4 studies excluded those with previous CVD. Is it also possible to provide a pooled estimate for these 4 studies?

6. PLOS authors have the option to publish the peer review history of their article (what does this mean?). If published, this will include your full peer review and any attached files.

Reviewer #1: No

Reviewer #2: Yes: Joline W Beulens

---

## [Author Response · Author response to Decision Letter 0]

19 Jun 2020

Dear Reviewers,

Re: PONE-D-20-11782 - Is breast arterial calcification associated with coronary artery disease? - A systematic review and meta-analysis

We would like to thank you for your helpful comments for improving our paper. We have addressed each as follows:

• Abstract:

o Updated with additional data

• Introduction:

o Reference made about BAC being a form of medial arterial calcification 

o Emphasised that our primary aim will add to current literature by briefly commenting on prior systematic reviews and meta-analyses 

Updated with more studies added

Our review will look at CAD across multiple modalities

• Methods:

o Provided further details on our search strategy

o Defined CAD in more detail

o Stratified studies based on whether BAC was quantified or not

o Added a sub-analysis of studies that quantified BAC and its association with the presence of CAD

• Results:

o Provided the 9 missing data on Table 1 that was left out during the initial editing process – does not affect results and conclusions

o Studies evaluating BAC and CAD and BAC and CAD sections

Number of studies have changed as we have decided to focus primarily on CAD and have excluded studies that had combination endpoints (e.g. CAD and stroke) – this has slightly increased the odds of BAC and CAD but has not changed our conclusions

Provided results regarding studies that have quantified BAC

• A new table has been included in the manuscript detailing the different methods used by studies to quantify BAC (Table 2)

• Association of BAC and CAD not affected by method of scoring BAC

• Sub-analysis showed moderate to severe BAC associated with CAD but not mild BAC using the two most commonly used method of assessing BAC severity (4 point and 12 point scale system) (Fig 3)

Stated the ORs of BAC and CAD in women asymptomatic of CAD.

o BAC and cardiovascular risk factors section

Fig 4 (previously Fig 3) has been modified to emphasise that data taken from baseline characteristics of cohort studies are cross-sectional data.

Small changes in numbers from the Schnatz paper has led to minor alterations in the pooled ORs. This has not affected overall results and conclusion (used prospective data rather than baseline data)

• Discussion:

o Made changes relevant to results

o New paragraph added to discuss results on the severity of BAC and the association of CAD

• Limitations:

o Added a statement regarding the use of primarily one researcher in data extraction and analysis as a limiting factor

• Conclusion: 

o Added comment that there may be an important association between BAC severity and CAD

• Supplementary material:

o Table S2-S4 has been modified to incorporate further details on how study quality was assessed

• Other analysis carried out but not included in manuscript:

o Stratified analysis of cross sectional and cohort studies

The following are our responses to each reviewer in greater detail:

REVIEWER ONE

1. The current meta-analysis does not add to the body of evidence concerning BAC and cardiovascular risk.

We thank the reviewer for this comment and have modified the manuscript to clearly state the additional information provided by our analysis regarding BAC and CAD. Three prior papers aimed to summarise the association between BAC and CVD. Abi Rafeh et al (2012) included only five studies and reported a positive OR of 1.59 (95%CI 1.21-2.09). Jiang et al included ten studies and summarised the odds of the association of BAC and CAD on CAG as 3.86 (95%CI 3.25-4.59). The most recent systematic review by Hendriks et al did not perform a meta-analysis but reported the range of adjusted hazard ratios as a range based on three identified studies (HR 1.32 (95%CI 1.08-1.60) to HR 1.44 (95%CI 1.02-2.05). All these prior papers included only the use of CAG and excluded studies that assessed CAD using CT-scans or questionnaires. It is recognised that CAG, being a luminogram, underestimates the presence of CAD in the presence of remodelling, which is common in early and even advanced atherosclerosis. CT-scans, on the other hand, can detect coronary plaque even when the coronary lumen is widely patent. We will be the first to systematically review studies that examined CAD from questionnaires, CAG, MPS and CTs. Our systematic review will also be the first to assess whether the association of BAC and CAD is sensitive to 1) the method used to assess CAD or 2) whether or not only patients are truly asymptomatic or have symptoms suggestive of CAD are included and 3) the method of reporting or grading BAC).

Only one prior study has evaluated the relationship between BAC and CVD risk factors. Hendriks et al looked at the associations of BAC and hypertension (11 studies), diabetes (14 studies), hypercholesterolemia (4 studies) and smoking (20 studies). Only studies of moderate to good quality were included in their analysis and amongst these, a positive association between BAC and diabetes and an inverse association between BAC and smoking was reported. We included studies of all quality and did a sub-analysis that included only good quality studies. This has resulted in a higher number of studies (hypertension: 42 studies; diabetes: 51 studies; hypercholesterolemia: 27 studies; smoking: 35 studies) included in our analysis but with multiple publications since then, we also had higher number of good quality studies in our sub-analysis (hypertension: 19 studies; diabetes: 21; hypercholesterolemia: 13; smoking: 20). By doing so, our results are in keeping with those by Hendriks et al, but we provided new evidence on the relationship between BAC and hypertension. This is probably due to the higher number of studies we have included in our analysis.

2. Search strategy should be explicated in Methods.

Thank you for your comment on our search strategy. We have expanded the information of our search strategy in our methodology under data sources and searches. To emphasise that our primary aim is to look at CAD specifically, we have defined CAD clearer in the study selection section of the methods. 

3. Even if this review was conducted in concordance with the PRISMA statement, ideally two researchers should independently perform the data extraction.

We have acknowledged this in our study limitations.

4. As description of method of quantifying BAC has been extracted, it would have been interesting to present results about that, that currently represent the crucial gap in clinical translation of BAC for preventive cardiology in women. No published systematic data are on this.

Thank you for this interesting point. We agree that this will be interesting and novel data and have performed a sensitivity analysis by stratifying studies that have graded BAC and those that have not. We found that there were no changes to the association nor heterogeneity. An apparently higher odds was observed in studies that graded the severity of BAC vs reporting only the presence or absence of BAC ((OR 4.04 (95%CI 2.59-6.30) vs OR 2.05 (95%CI 1.70-2.47) and this is likely due to population differences but could also indicate that semi-quantitative methods are more sensitive at detecting subtle forms of BAC. 

We identified 12 studies that have visually graded the severity of BAC and these are summarised in a table (Table 2). Of these, five studies used the same 4 point scale system and 3 studies used the same 12 point scale system. The other 4 studies have used different grading systems that could not be grouped together. There were only 9 studies with extractable data (4 studies with the 4 point scale system; 3 studies with the 12 point scale system; 2 studies that could not be grouped). We chose to perform two sub-analyses on BAC severity and the association of CAD using studies from 4- and 12-point scale system. These were stratified to mild and moderate/severe BAC. We chose to group moderate and severe BAC together due to the small numbers. Interestingly, we found that a positive association between moderate to severe BAC (OR 4.83 (95%CI 1.50-15.54) and OR 2.95 (95%CI 1.49-5.84), respectively) but not mild BAC (OR 2.04 (95%CI 0.82-5.05) and OR 1.08 (95%CI 0.42-2.75), respectively). We believe that this suggest a possible association between BAC severity and CAD. This advocates for the need of a consensus in a method of assessing BAC severity and also larger prospective studies which incorporate grading of BAC. Furthermore, all studies to date have used a semi-quantitative method that is subjected to biasness and we have suggested that a direct quantitative method may be warranted. We have included this analysis in the results and added a discussion on the implications.

5. What’s the BAC prevalence estimate?

We have provided the prevalence of BAC in Table 1 to allow appreciation of BAC prevalence amongst different population groups. In the general population, BAC prevalence varied from 3.0 to 63.7% with notably higher BAC prevalence in those who were symptomatic for CAD. In post-menopausal women, BAC prevalence ranged from 8.5 to 59.5%. BAC prevalence seemed to be higher in women with diabetes and chronic kidney disease at 36.5 to 45.1% and 14.5 to 54.8%, respectively. 

6. Table 1 report only 50 studies. Are 9 studies missing?

Thank you for pointing out this error. Whilst all studies were included in the generation of the results, there were 9 studies that were accidentally removed from Table 1 during the editing process. This has been corrected. Importantly, the data from these studies were included in our initial analysis so that the relevant results and conclusions have not changed.

7. What’s the EMBED database? Do you mean EMBASE? Otherwise please give source.

Yes, we meant EMBASE and this has been corrected in the manuscript.

REVIEWER 2

1. Introduction: no mention is made of the fact that BAC concerns a specific form of calcification namely medial calcification and that this form of calcification may also have different pathways to lead to CVD. This should be indicated in the paper.

We recognise the importance of highlighting that BAC is a form of medial arterial calcification and could explain why it is not routinely reported. We have now discussed this in our introduction. 

2. Introduction: an earlier systematic review and meta-analysis was performed on this topic (Hendriks EJ, atherosclerosis, 2015). This study should be referenced and the authors should indicate what this additional meta-analysis will add.

Thank you for your comment. We believe that this will further highlight the importance of doing an updated systematic review on the association of BAC and CAD as the prior systematic review had only looked at three longitudinal studies for the association of BAC and cardiovascular events. One of them looked at cardiovascular mortality and the other two had information on the association of BAC and CAD. Our systematic review has specifically examined the association between BAC and CAD and is the first systematic review to include all studies (case controls, cross sectional, cohort studies) in our analysis. This has added an additional 29 studies to the previous systematic review (total 31 studies that examined BAC and specifically CAD). Furthermore we also include studies that have assessed the presence of CAD using CT-scans or questionnaires, enabling a further sensitivity analysis with discussion of the strengths and limitations of using these different methodologies.

3. Methods; only 1 reviewer performed study selection, data extraction and quality assessment. This is a limitation of the study.

We have acknowledged this in our study limitations.

4. Methods: can the authors provide a little more detail on how quality was assessed for study population, control for confounding and the outcome. The number of points is indicated but not how this is specifically applied to these studies and which items under these headings were scored. This information could be provided in a supplement.

The tables regarding the scoring of each study has been modified to include further details on how quality was assessed (see S2-S4 Tables).

5. Methods; CAD is very heterogenous endpoint. Can the authors perform subgroup analyses for different outcomes such as coronary heart disease and stroke or otherwise specify whether only events or also CAD mortality was investigated and perform subgroup analyses for this.

Thank you for your constructive feedback. We agree that cardiovascular disease is a very heterogenous endpoint and may include CAD and stroke. As such, we have decided to examine the association of BAC and specifically CAD. To facilitate this, the definition of CAD has now been included where CAD is defined by medical history/records (history of myocardial infarction or angina, positive stress test or abnormal CAG, coronary revascularisation), presence of CAD on CAG, presence of perfusion defect on MPS and presence of coronary artery calcium on CTCA, CT chest and CCS. As a result, we have re-reviewed all studies that were included in our initial analysis and found that there were three studies (Kelly, Soran, Ferreira) that combined multiple endpoints as cardiovascular disease (e.g. stroke, thrombosis, transient ischaemic attacks). As it is not possible to extract data specifically on CAD from these studies, we have now excluded them. We also had to change the data entered from the Schnatz paper as there were multiple endpoints for CAD which included abnormal CAG and history of myocardial infarction. The initial analysis extracted data from history of myocardial infarction but we feel that abnormal coronary CAG may be a better reflection of the presence of CAD. We have made a point of this in our methods (line 207-209). We have also switched the Kemmeren paper with the van Noord paper as the latter has listed the specific number of participants with a history of myocardial infarction. Both these papers were from the same cohort but the Kemmeren paper had combined the endpoints of history of myocardial infarction, stroke/TIA and thrombosis. As such, the number of cohort studies have reduced from six to three. Therefore, the final number of studies included in our analysis is now 31. Despite these changes, our conclusions have not changed and the OR for the association of BAC and CAD has now increased from 2.56 (95%CI 2.11-3.12) to 2.61 (95%CI 2.12-3.21). All sub-analysis that were performed in the initial analysis were also repeated with the same conclusions. 

With regards to CAD mortality, there is only one study that have looked at the association of BAC and CAD mortality – Kemmeren et al at where an adjusted HR of 1.44 (95%CI 1.02-2.05) was observed. We have not included this study in our analysis as we have now chosen to specifically look at CAD. This is reflected by the changes on Table 1 which now excludes the Kemmeren paper and included the van Noord paper. 

6. Results: can the authors also do subgroup analysis separating cross-sectional from prospective studies?

A subgroup analysis separating cross-sectional from prospective studies had no impact on the odds as well as heterogeneity. The pooled ORs were 2.68 (95%CI 2.11-3.40) and 2.53 (95%CI 1.43-4.45) for cross sectional and prospective studies, respectively. We have included these results in the supplementary materials.

7. Discussion: the authors indicate that only 4 studies excluded those with previous CVD. Is it also possible to provide a pooled estimate for these 4 studies?

The pooled estimate for these 4 studies remained positive at a higher odds of 3.46 (95%CI 1.57-7.61) with significant heterogeneity at I2=84% (p<0.01). We will report this in our results (line 329-331) and discussion (line 490).

We would like to thank both the Reviewers and Editors for your constructive comments, which we feel has improved our manuscript.

With many thanks,

Dr Sing Ching Lee

---

## [Decision Letter · Decision Letter 1]

10 Jul 2020

Is breast arterial calcification associated with coronary artery disease? - A systematic review and meta-analysis

PONE-D-20-11782R1

Dear Dr. Lee,

We’re pleased to inform you that your manuscript has been judged scientifically suitable for publication and will be formally accepted for publication once it meets all outstanding technical requirements.

Kind regards,

Pascal A. T. Baltzer, M.D.

Academic Editor

PLOS ONE

Additional Editor Comments (optional):

Reviewers' comments:

Reviewer's Responses to Questions

**Comments to the Author**

1. If the authors have adequately addressed your comments raised in a previous round of review and you feel that this manuscript is now acceptable for publication, you may indicate that here to bypass the “Comments to the Author” section, enter your conflict of interest statement in the “Confidential to Editor” section, and submit your "Accept" recommendation.

Reviewer #1: All comments have been addressed

Reviewer #2: All comments have been addressed

2. Is the manuscript technically sound, and do the data support the conclusions?

Reviewer #1: Yes

Reviewer #2: Yes

3. Has the statistical analysis been performed appropriately and rigorously? 

Reviewer #1: Yes

Reviewer #2: Yes

4. Have the authors made all data underlying the findings in their manuscript fully available?

Reviewer #1: Yes

Reviewer #2: Yes

5. Is the manuscript presented in an intelligible fashion and written in standard English?

Reviewer #1: Yes

Reviewer #2: Yes

6. Review Comments to the Author

Reviewer #1: Thank you for addressing all my points. An original and relevant section part have been included. v

Reviewer #2: (No Response)

7. PLOS authors have the option to publish the peer review history of their article (what does this mean?). If published, this will include your full peer review and any attached files.

Reviewer #1: No

Reviewer #2: **Yes: **Joline WJ Beulens

---

## [Editor Report · Acceptance letter]

16 Jul 2020

PONE-D-20-11782R1 

Is breast arterial calcification associated with coronary artery disease? - A systematic review and meta-analysis 

Dear Dr. Lee:

I'm pleased to inform you that your manuscript has been deemed suitable for publication in PLOS ONE. Congratulations! Your manuscript is now with our production department. 

Kind regards, 

on behalf of

Dr. Pascal A. T. Baltzer 

Academic Editor

PLOS ONE